# Functions and Mechanisms of Pro-Lysyl Oxidase Processing in Cancers and Eye Pathologies with a Focus on Diabetic Retinopathy

**DOI:** 10.3390/ijms23095088

**Published:** 2022-05-03

**Authors:** Philip C. Trackman, Yaser Peymanfar, Sayon Roy

**Affiliations:** 1The Forsyth Institute, 245 First Street, Cambridge, MA 02142, USA; ypeymanfar@forsyth.org; 2Department of Translational Dental Medicine, Boston University Henry M Goldman School of Dental Medicine, 700 Albany Street, Boston, MA 02118, USA; 3Department of Medicine, Boston University School of Medicine, 650 Albany Street, Boston, MA 02118, USA

**Keywords:** lysyl oxidase, lysyl oxidase propeptide, structure–function, diabetes, eye pathologies

## Abstract

Lysyl oxidases are multifunctional proteins derived from five lysyl oxidase paralogues *(LOX)* and lysyl oxidase-like 1 through lysyl oxidase-like 4 (*LOXL1–LOXL4*). All participate in the biosynthesis of and maturation of connective tissues by catalyzing the oxidative deamination of lysine residues in collagens and elastin, which ultimately results in the development of cross-links required to function. In addition, the five *LOX* genes have been linked to fibrosis and cancer when overexpressed, while tumor suppression by the propeptide derived from pro-LOX has been documented. Similarly, in diabetic retinopathy, LOX overexpression, activity, and elevated LOX propeptide have been documented. The proteolytic processing of pro-forms of the respective proteins is beginning to draw attention as the resultant peptides appear to exhibit their own biological activities. In this review we focus on the *LOX* paralogue, and what is known regarding its extracellular biosynthetic processing and the still incomplete knowledge regarding the activities and mechanisms of the released lysyl oxidase propeptide (LOX-PP). In addition, a summary of the roles of both LOX and LOX-PP in diabetic retinopathy, and brief mentions of the roles for LOX and closely related LOXL1 in glaucoma, and keratoconus, respectively, are included.

## 1. Introduction

Lysyl oxidases are encoded by a family of five paralogue genes known as lysyl oxidase (*LOX*) and lysyl oxidase like-1, -2, -3, and -4 (*LOXL1*–*LOXL4*). All are synthesized as pro-proteins which are structurally conserved in vertebrates [1], with LOX and LOXL1 being more closely related to each other [2] than to LOXL2–LOXL4 [3]. These pro-proteins contain a conserved C-terminal enzyme domain, while the propeptide regions of LOX and LOXL1 are each largely unique in sequence [4]. By contrast, LOXL2–LOXL4 propeptides are more similar to each other and all three contain four conserved Scavenger Receptor Cysteine-Rich (SRCR) domains, which typically serve as protein–protein binding domains in other proteins [5,6]. LOX is processed and activated by cleavage of its N-terminal pro-peptide catalyzed principally by Bone Morphogenetic Protein-1 (BMP-1) or BMP-1-related procollagen C-proteinases Tolloid Like-1 and -2 following secretion into the extracellular environment [7]. LOXL1 is similarly processed by BMP1 [8], while LOXL2 undergoes processing by furin enzymes required for type IV collagen crosslinking [9,10].

The lysyl oxidase family proteins all participate in the biosynthetic maturation of collagens and elastin by catalyzing the extracellular oxidation of the ε-amino group of peptidyl lysine or peptidyl hydroxy lysine residues in procollagens and tropoelastin to form the corresponding aldehydes, which are required for the subsequent formation of the biosynthetic cross-links and the function of the extracellular matrix [4]. In tumor biology, in general, abnormal high levels of lysyl oxidase paralogues expressed by tumor cells and/or associated stromal cells correlate well with poor outcomes in cancer. By contrast, the unique propeptide derived from pro-LOX (LOX-PP) has tumor growth inhibitory properties consistent with being a tumor suppressor [11]. In this review we will focus on findings specifically regarding the properties and mechanisms of pro-LOX processing and LOX-PP in the context of ECM modulation, cancers, and diabetic retinopathy. Brief mentions of LOX and LOXL1 relevance to some additional eye pathologies are made here that will be more extensively discussed in other articles in this special issue of IJMS.

## 2. LOX-PP Is Not Trash: Its Role in Tumor Suppression

The notion that LOX-PP, released extracellularly during the biosynthesis of the active LOX enzyme (Figure 1), could have its own function came from the observation that the tumor suppressor function linked to the phenotypic reversion of *c*-H-*ras* transformed fibroblasts depended in some way on the restoration of LOX expression [12,13]. At the time it was assumed that LOX enzyme activity fulfilled this function. However, treatment of phenotypically reverted *c*-H-*ras* transformed fibroblasts with the selective suicide substrate inhibitor of LOX, β-aminopripiontrile (BAPN), failed to re-transform these cells, and failed to induce a transformed phenotype [14]. This finding led to the hypothesis that LOX-PP, produced stoichiometrically with the LOX enzyme, could promote phenotypic reversion, which was ultimately demonstrated [14]. Subsequent follow-up studies identified several signaling pathways and binding partners inhibited by LOX-PP which mediated *ras*-effectors signaling, and inhibited xenograft growth in mice. Subsequent studies demonstrated that the anti-tumor activity of LOX-PP is through the direct or indirect inhibition of the Hsp70 and the suppresion of the MAPK/ERK pathway [13,15], Akt, FGFRs, [16,17] RPTPĸ signaling [18], FAK [19], and others [20] depending on the cellular or cancer context [4,21]. LOX-PP can inhibit tumorigenesis by reducing the vascular endothelial growth factor (VEGF) in human umbilical vein endothelial cells and the suppression of vascular tube formation in chick chorioallantoic membranes [22,23]. Induction of LOX-PP expression by the adenoviral vector reduced cancer cell migration and hampered the expression of angiogenic factors MMP2 and MMP9 [24]. LOX-PP was shown to have an interaction with EGF and the tumor endothelial marker-8 (TEM-8) on the surface of activated endothelial cells and to possibly control angiogenesis [21].

LOX-PP is generated extracellularly by the proteolytic processing of pro-lysyl oxidase at Gly168/Asp169 by procollagen C-proteinases and is likely to target FGFRs and other cell surface receptors [17,21] (Figure 1). However, LOX-PP also can re-enter cells in order to interact with its intracellular targets. A major mechanism of LOX-PP uptake is by macropinocytosis that in most cases is a receptor-independent mode of endocytosis [25]. LOX-PP has a very high pI in part due to its high content (12%) of arginine residues, and evidence indicates that LOX-PP in endosomes increases the internal pH of endosomes, which may lead to cytoplasmic release permitting the LOX-PP targeting of intracellular cytoplasmic signaling molecules, rather than the simple trafficking of LOX-PP to lysozomes for degradation [25].

A new question then becomes how can LOX-PP inhibit so many targets, and which target(s) is (are) most important for its tumor inhibitory function? Structure prediction tools have revealed that LOX-PP has a mostly intrinsically disordered structure, which is consistent with its ability to target multiple proteins [26]. Intrinsically disordered proteins adopt biologically active conformations when interacting with targets. Some disordered proteins have multiple functional binding partners [27], which is also the case for LOX-PP.

### 2.1. LOX-PP Regions That Mediate Its Function

Detailed structure/function studies employing site-directed mutagenesis or deletion mutants and a naturally occurring polymorphism have revealed different direct targets involving different LOX-PP subregions or functional amino acid sequences, as can be expected for an intrinsically disordered protein. Relevant sequences for most LOX-PP targets have not yet been mapped to specific LOX-PP amino acid sequences but are not assumed to be less important than those that have been mapped. The mapped LOX-PP sequences that interact with other proteins are summarized below and are also illustrated in Figure 2.

#### 2.1.1. Intracellular Functional Binding Partners of LOX-PP

The signaling adapter CIN85 protein and its binding partner c-CBL function to promote cancer cell invasiveness. Their additional binding partners have roles in regulating cytoskeletal and membrane structures [28,29]. CIN85 has been implicated in the control of levels of NADPH oxidases and reactive oxygen species in cancer cell lines [30]. CIN85 has three SH3 domains: A, B, and C, respectively. LOX-PP was shown to bind only to the SH3 B domain and not to A or C in breast cancer cell lines [28]. This binding occurs via conserved human LOX-PP amino acids 117–123 (mouse and rat 111–117) via its PxPxxR sequence instead of the more typical PxxPxR SH3 binding consensus sequence (Figure 2). The LOX-PP binding interaction with CIN85 at this site inhibits the invasive phenotype of breast cancer cells on Matrigel and the collagenolytic activity of transfected cells plated on a collagen substrate [28]. These findings point to one mechanism by which LOX-PP can inhibit the invasiveness of cancer cells in its anti-metastatic activity.

*Hsp70 and c-Raf*. Pull-down studies in HEK293T cells revealed that LOX-PP interacts with Hsp70, and c-Raf. Binding to α,β-tubulin was also confirmed [28,31]. The direct binding sequences of LOX-PP to these proteins were determined by coprecipitation studies of bacterially expressed wild type LOX-PP subregions and the residues 26–110 of rat LOX-PP (human 32–116) were identified as the peptide regions that bind to Hsp70, c-Raf, and α-tubulin. These regions are distinct from the CIN85 binding site. The precise respective binding site amino acid residues in this peptide region to Hsp70, cRaf, α,β-tubulin were, however, not identified. Thus, this rat peptide 26–110 (human 32–116) may contain different regions for the three different proteins. It may be of interest that human residues 39–66 of LOX-PP are conserved between LOX and LOXL1 [32], and that this sequence has been predicted to be structured, unlike most of LOX-PP that is intrinsically disordered [21] (Figure 2). LOX-PP–Hsp70–cRaf interactions resulted in reduced Erk1/2 signaling, which is Ras/c-Raf pathway-dependent, and consistent with the tumor growth inhibitory properties of LOX-PP. The attenuated chaperone function of Hsp70 by LOX-PP was also identified [28]. The biological consequences of LOX-PP binding to tubulin have not been determined to our knowledge.

#### 2.1.2. LOX-PP Polymorphism

A polymorphism resulting in a human point mutation of amino acid residue 158 from Arg to Gln (rs1800449), equivalent to residues R152 in rat and mouse LOX-PP, results in increased cancer susceptibility in a variety of cancers. This residue is not contained in the binding sites summarized above. A knockin mouse mimicking the human polymorphic variant has been made and was shown to be more susceptible to chemically induced breast cancer [33]. Ongoing unpublished studies have indicated that this was also true in a model of chemically induced oral cancer in knockin mice harboring this mutation. The clinical importance of rs1800449 to increase the incidence or severity of cancers is supported by several independently published human epidemiology studies [34,35,36,37,38,39,40]. The identification of this polymorphic variant with reduced anti-tumor activity may offer the opportunity to further understand what the “normal” Arg158 polymorphic variant targets that the “mutant” Gln158 variant does not. Alternatively, the Gln158 variant may function by a dominant negative mechanism. In this regard it is of interest that MMP2 can cleave to pro-LOX at residue 156N/157L, immediately adjacent to Arg158 in the predominant form of Pro-LOX, or potentially Gln in the less common form [22,41,42] (Figure 2). “Normal” BMP1/procollagen C-proteinase cleavage occurs at human LOX residue 168/169 (mouse residue 162, Figure 2). It seems possible that the polymorphic Gln variant could be resistant to MMP2 proteolytic cleavage at residue 158, favors the “normal” processing only by procollagen C-proteinases at 168, resulting in active lysyl oxidase that is well-described, while the MMP2 cleavage product at 158 could have an altered activity or substrate specificity. In this regard, MMP2 is well-known to be upregulated in a variety of cancers, suggesting a possible post-translational control of altered LOX enzyme activity or specificity in a cancerous microenvironment. It is similarly unclear if the shorter or longer produced LOX-PPs or short peptide fragment 159–167 would have different tumor modulating properties as might be expected from the apparent loss of tumor inhibitory properties of the Gln variant.

#### 2.1.3. LOX-PP Gln Variant Fails to Inhibit FGFR1

LOX-PP has been shown to inhibit FGF signaling in prostate cancer cell lines with a strong inhibition of proliferation and primarily targeted Akt signaling, and Erk1/2 to a smaller degree. These effects were shown to be mediated by FGFR1 using neutralizing FRFR1 antibodies [16]. In the phenotypically normal MC3T3-E1 cell line, LOX-PP similarly inhibited cell proliferation, by targeting primarily Erk1/2 and not Akt. Binding kinetics of LOX-PP to FGFR1 suggested that LOX-PP binds directly to FGFR1, but to a site other than the FGF2 binding site [17]. In murine breast cancer cells, LOX-PP similarly inhibited FGF2 stimulated proliferation and signaling in vitro and xenograft growth in mice, while the Arg152Gln rat variant did not inhibit signaling and inhibited xenograft growth to a lesser degree than Arg152 LOX-PP [43]. More recently, an interesting study in 3T3-L1 cells found that LOX-PP increased expression of PPARγ and C/EBP and therefore adipogenesis, inhibited both Akt and Erk1/2 signaling, all mimicked by an unrelated FGFR inhibitor. The mouse Arg152Gln variant of LOX-PP failed to both stimulate adipogenesis, PPARγ and C/EBP expression [42]. These independent findings all point to FGFR-mediated signaling as relevant to both the biological activity of LOX-PP in different contexts, and to the importance of understanding more about the mechanistic and biological consequences of the Arg/Gln polymorphism in a variety of physiological contexts [43,44,45].

### 2.2. ADAMTS2/14 Processing of Pro-LOX

An additional proteolytic processing site in human pro-LOX with functional consequences was recently identified by Rosell-Garcia and colleagues [46]. A site downstream of the procollagen processing site (human residue 167/168) found at 218/219 results in a longer propeptide and shorter (25 kDa vs. ~30 kDa) active enzyme. The resulting 25 kDa enzyme was found to retain its activity against low molecular weight amines. By contrast, on a solid phase binding assay the 25 kDa isoform exhibited a reduced ability to bind collagen that contained telopeptides. The telopeptides contain the lysine residues oxidized by LOX, so reduced binding to this entity strongly implies a reduced ability of the shorter LOX enzyme to function as a collagen cross-linking catalyst. The sequence between 168 and 218 is rich in tyrosine and the data indicated that some of these tyrosine residues are sulfated in the longer BMP1/procollagen C-proteinase processed LOX-PP, possibly facilitating collagen binding of the resulting mature 30 kDa enzyme [46]. Since the metastatic activity of LOX is strongly linked to its enzyme activity and fibrosis, one could propose that high levels of ADAMTS2 or 14 would act as fibrosis inhibitors or tumor suppressors. However, ADAMTS 2 and 14 are procollagen N-proteinases required for the N-terminal biosynthetic processing of types I, II, III and V procollagens, and several additional ECM molecules [47,48]. Interestingly, loss of ADAMTS2 results in poor skin collagen structure in the condition known as dermatosparaxis in animals and humans [49], consistent with low levels of N-terminal pro-collagen N-terminal propeptide processing, and low collagen cross-linking, as would be expected. By contrast, ADAMTS2 overexpression has been linked to an increased incidence of some forms of gastric [50] and oral cancers [51,52], possibly suggesting that longer LOX-PP isoforms containing sulfated tyrosine residues resulting from ADAMTS 2 or 14 cleavage only, should they accumulate, may reduce the tumor suppressor functions of LOX-PP and/or increase LOX enzyme tumor promoting effects. One possibility is that the 25 kDa enzyme generated after ADAMTS2/14 cleavage would have increased substrate specificity for a tumor promoting target rather than collagen, such as PDGFRβ, increasing PDGF signaling and cancer progression [53].

## 3. LOX, LOX-PP and Eye Pathologies

This section focusses on diabetic retinopathy and two additional prominent ocular diseases associated with abnormal LOX family and LOX-PP activities. It should be noted that LOX and LOX-PP have been reported to be involved in other eye diseases which are not discussed here as it is beyond the scope of this review.

### 3.1. Diabetic Retinopathy

The rising incidence of diabetes has led to a dramatic increase in diabetic complications worldwide. Despite the introduction of treatment strategies, diabetic retinopathy (DR) remains a major cause of blindness and is one of the most common microvascular complications of diabetes [54,55,56,57] for which no preventive therapy is currently available. A significant clinical manifestation of DR is retinal vascular cell death and the breakdown of the blood retinal barrier (BRB) leading to excess vascular leakage and macular edema [58]. The BRB is compromised by the thickening of the retinal capillary BM, a hallmark of DR [59]. The biogenesis and maturation of the vascular BM is dependent on LOX-mediated cross-linking of ECM components. Aberrant LOX activity results in abnormal post-translational modification of BM collagens, affects its functionality, and predisposes connective tissues to certain diseases [60,61,62,63]. Several studies have reported elevated LOX activity and increased collagen cross-linking in diabetic tissues such as the lungs of diabetic rats [64] and the skin of diabetic patients where elevated LOX activity correlated with duration of diabetes, glycemic control, and long-term complications [65]. By contrast, abnormally low biosynthetic collagen cross-linking and low lysyl oxidase levels occur in diabetic bone [66,67], pointing to different functional regulatory pathways in different tissues. To add to this complexity, a recent study indicated that LOX upregulation may contribute to ECM production independent of its crosslinking function [68]. This raises the possibility that LOX upregulation in diabetes could promote vascular BM thickening and play a pathogenic role in DR by both enzymatic and non-enzymatic mechanisms. Interestingly, LOX propeptide (LOX-PP), which has no LOX enzyme activity, released during LOX processing, was recently shown to promote apoptosis in various diseased tissues, including diabetic retinas [24,69,70,71]. These reports raise the possibility that a high glucose condition increases LOX and LOX-PP levels, which in turn promote apoptosis and increased retinal vascular permeability [71,72,73,74,75]. By contrast, a study reported decreased LOX activity in the vitreous of eyes with proliferative diabetic retinopathy [76]. In this study, vitreous samples obtained from patients were compared for LOX mRNA levels and specific activity of LOX, MMP-2 and -9 with those obtained from autopsy eyes, while a different study reported increased LOX mRNA levels in human retinal pigmented cells exposed to high glucose. Ongoing studies are investigating possible links between abnormal LOX and LOX-PP levels and the development of retinal vascular cell loss and capillary leakage, and whether a strategy targeting LOX and LOX-PP could be effective in preventing vascular lesions in DR.

### 3.2. Keratoconus

Keratoconus (KC) is a corneal disease characterized by progressive central cornea thinning and conical protrusion of the cornea. The cellular mechanisms underlying the development and progression of the disease remain unclear. Morphological changes in different structures of KC corneas including epithelium, basement membrane, nerve fibers, Bowman’s layer, stroma, Descemet membrane, and endothelium have been identified. Additionally, the diversity of the structures appear to represent temporal differences during disease progression [77]. Other observations support the involvement of several biochemical events that regulate cellular and extracellular processes, proliferation, differentiation, and apoptosis of keratocytes, and oxidative damage [78]. The disease can be diagnosed by assessing clinical signs such as stromal thinning, together with accurate computer-assisted video-keratography. KC has a prevalence of approximately 1:2000.

The involvement of LOX has been identified during the development of KC. In particular, a number of studies have reported a lower expression and reduced activity of LOX and/or cross-linking defects in corneas of individuals with KC [79,80,81]. Decreased expression of LOX appears to contribute to the structural deformity of the KC cornea [80]. The risk of KC development may be associated with specific LOX polymorphisms [82]. Moreover, abnormal LOX activity is a risk factor for KC, and genetic evidence indicates LOX variants likely increase susceptibility to developing KC [83]. Furthermore, LOX is differentially up-regulated in corneal epithelial cells of KC patients compared to those in healthy controls [84]. The structural deformity of the KC cornea may be related to unbalanced expressions of collagens (no change) and LOX (upregulated), and elevated MMP9 in the corneal epithelium. In recent years, a number of studies have revealed inflammatory cytokines and matrix-remodeling enzymes to participate in the development of KC [85].

Family-based and case control studies have revealed that variations in the LOX gene could increase susceptibility of KC development. Interestingly, mutational screening of LOX in a cohort of 225 sporadic and 77 familial KC cases showed no involvement of LOX with KC [86]. Genetic variants in additional genes may interact with changes in LOX. It is notable that a novel gene for KC has been localized to a 5.6-Mb interval on 13q32 [87]. The 13q location is notable because all known genes at this locus were not mutated and therefore excluded as functionally relevant to KC. How this locus is functionally relevant to KC remains unknown, to our knowledge. Although there is increasing evidence that LOX is involved in the pathogenesis and progression of KC, further studies are needed to dissect LOX’s role including other factors and gain deeper insight into the molecular mechanisms underlying KC development [83].

### 3.3. Glaucoma

Among patients with primary open angle glaucoma (POAG), the prevalence of pseudoexfoliation appears to be linked to geographic variation. Lysyl oxidase-like 1 (*LOXL1*) gene polymorphisms have been widely studied in different ethnic populations. The literature suggests conflicting reports related to *LOXL1* gene variants and glaucoma, in particular, related to pseudoexfoliation and primary open angle glaucoma. A recent study of the north Indian population reported a lack of association between the *LOXL1* gene polymorphisms and primary open angle glaucoma [88] consistent with a study that suggests that *LOXL1* polymorphisms are not associated with POAG risk, based on meta-analysis [89]. However, a prevalence of pseudoexfoliation glaucoma risk was reported to be associated with variants of the *LOXL1* gene in an Irish population [90], supported by additional studies indicating an association of *LOXL1* gene polymorphisms and POAG in Turkish patients [91,92], Spanish population [93], Greek patients [94], and German patients [95]. Further population-based studies on a large-scale basis are necessary to identify the worldwide distribution of pseudoexfoliation and primary open angle glaucomatous development and the relationship with *LOXL1*.

To gain insights into the role of rare *LOXL1* variants among different ethnic groups, transcriptome analyses could reveal further information and identify specific pathways and their associations with *LOXL1*. *LOXL1* remains a potential target to better understand the pathophysiology of the abnormal matrix, in particular, interactions with matrix components in relation to genetic factors present in ethnic populations. Ultimately, modulation of *LOXL1* gene expression could be promising and present a potential strategy for treatment of pesudoexfoliation and primary open angle glaucoma.

## 4. Conclusions and Perspectives

It is increasingly clear that the lysyl oxidase family of proteins derived from all five paralogues is multifunctional. With recent insights into new proteolytic processing sites of only pro-LOX, and the still limited structure/function studies of LOX-PP and additional derived peptides, there is a need to further identify and understand intracellular and extracellular mechanisms by which the active enzyme and enzymatically inactive products derived from pro-lysyl oxidase have function in tumor biology, fibrosis, DR and other ocular complications. Similar studies investigating the other four *LOX* paralogues would be of interest as they may reveal additional dimensions to the biology of this important gene family.

## Figures and Tables

**Figure 1 ijms-23-05088-f001:**
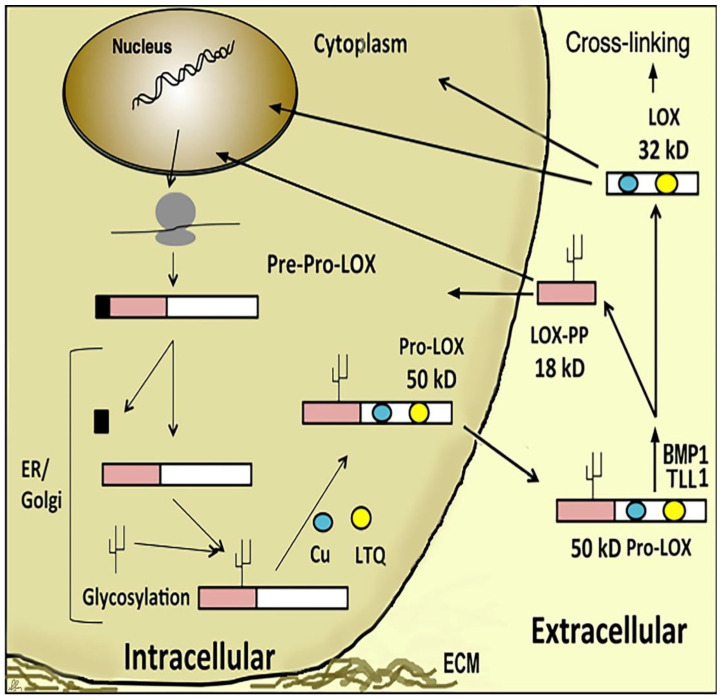
Diagram illustrating LOX and LOX-PP biogenesis. LOX is synthesized as a 50 kD inactive Pro-LOX and processed extracellularly by proteolytic cleavage into a functional 32 kD LOX enzyme and an 18 kD propeptide (LOX-PP). Additional proteolytic processing sites of Pro-LOX have recently been identified. LOX-PP has both intracellular and extracellular targets.

**Figure 2 ijms-23-05088-f002:**
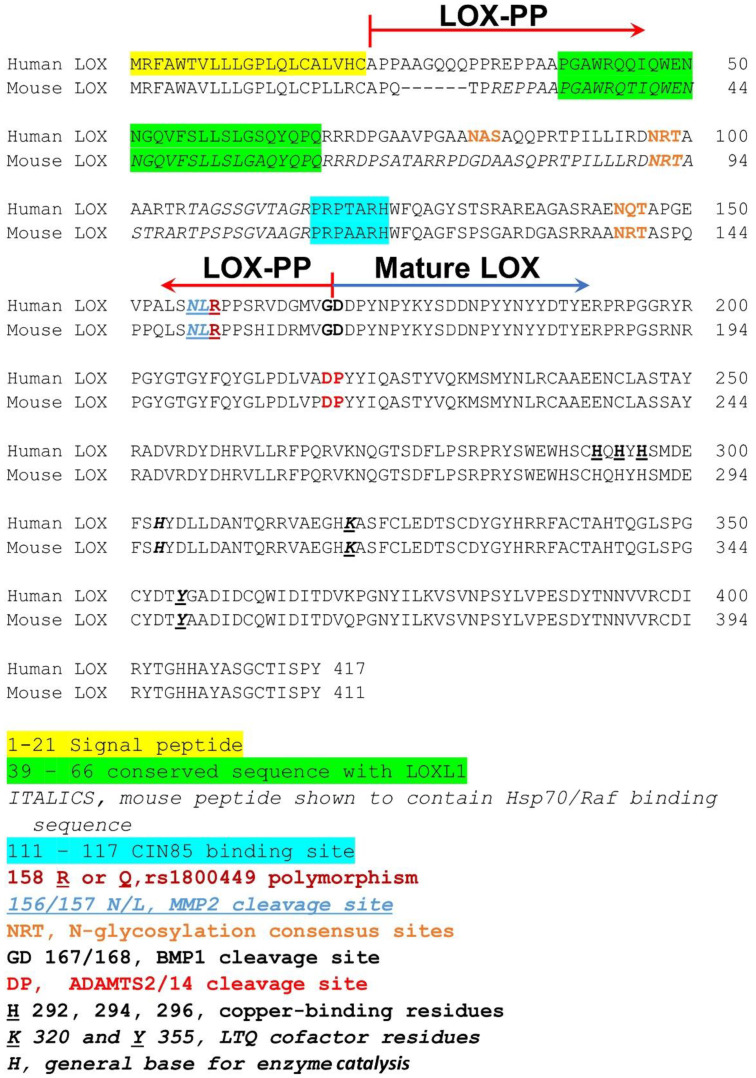
Aligned amino sequences of human and mouse pro-LOX, processing sites, active domains, and binding sequences. Symbols and colors are defined at the bottom of the figure and identify the sequences described in the text.

## Data Availability

Not applicable.

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
