# Peer review of "Functions and Mechanisms of Pro-Lysyl Oxidase Processing in Cancers and Eye Pathologies with a Focus on Diabetic Retinopathy"

_ijms, 2022, doi:10.3390/ijms23095088_

Round 1

Reviewer 1 Report

In their review, Trackman et al provide a comprehensive review on the role of lysyl oxidase (LOX) and its propeptide (LOX-PP) in cancers, diabetic retinopathy and keratoconus. In addition, they briefly discuss the role of LOXL1 in glaucoma. The review is very well written and mainly requires very minor editing. A few minor comments:

  • While most of the review focuses on the catalytic vs. non-catalytic function of LOX and LOX-PP, respectively, the sections on Keratoconus and glaucoma seem less clear with no mention at all of the pro-peptide.
  • Lines 271-272, “LOX is differentially expressed in corneal epithelial cells …,” is it upregulated or downregulated?
  • Line 280-281, “It is notable that a novel gene … on 13q32.” Why is this notable? Is 13q32 where LOX is?
  • Try to be consistent with the nomenclature: LOX = human, mouse or rat protein, LOX = human gene, Lox = mouse or rat gene.
  • Line 66 (near the end), delete “a” before “several.”
  • Line 106 (near the end), delet “and” before “but.”
  • Line 256, delete “The” from the beginning of the sentence “The morphological…”
  • Line 298, delete the word “respectively.”

Author Response

Reviewer 1

Comment: While most of the review focuses on the catalytic vs. non-catalytic function of LOX and LOX-PP, respectively, the sections on Keratoconus and glaucoma seem less clear with no mention at all of the pro-peptide.

Response: Keratoconus and glaucoma have received considerable attention in the literature since association of SNPs in LOX and LOXL1 have been linked or investigated as possibly contributing to these eye conditions. In addition to the work presented in diabetic retinopathy which is directly relevant to LOX-PP and LOX, we felt that the review would not be complete without acknowledging and briefly summarizing the literature on keratoconus and glaucoma for the benefit of those readers who may have a general interest in connections between eye pathologies and the lysyl oxidase family of genes and proteins. In this regard we have edited the title of the article to reflect our intent.

Comment: Line 280-281, “It is notable that a novel gene … on 13q32.” Why is this notable? Is 13q32 where LOX is?

Response: The Lox gene is located on chromosome 5, not on chromosome 13.  The 13q location is notable because the function of the locus is unknown, while all known genes at this locus were not mutated and therefore excluded . This has been clarified in the text.

Comment: Try to be consistent with the nomenclature: LOX = human, mouse or rat protein, LOX = human gene, Lox = mouse or rat gene.

Response: We have gone through the manuscript including Figure 2 and corrected this nomenclature.

Comment: Line 66 (near the end), delete “a” before “several.” Line 106 (near the end), delet “and” before “but.” Line 256, delete “The” from the beginning of the sentence “The morphological…” Line 298, delete the word “respectively.”

Response: Line 66, 106, 256 and 298 have been corrected as requested.

Reviewer 2 Report

This is a comprehensive review on the function of LOX propeptide (LOX-PP) and its relation to human diseases, although chapters 3.2 and 3.3 does not seem to be related to LOX-PP. Minor revision including following points would improve the manuscript.

  1. In Abstract, italicized LOX and LOXL1 - 4 means the human genes, whereas in Introduction, these genes are described as italicized Lox and Loxl1 - 4 that means mouse genes.
  2. Line 36, "in other proteins, [5,6]": this comma is not necessary.
  3. Line 60, "Lox expression" should be "LOX expression".
  4. Line 113, The title of 2.1.1. should be changed because this chapter contains not only CIN85, but also Hsp70 and c-Raf. More broad terms, e.g., "Intracellular target of LOX-PP" may be suitable for the title.
  5. Line 117, "NAPDH" should be "NADPH".
  6. Line 132, "Hsp70, cRaf, α, β tubulin and were" should be " Hsp70, cRaf, and α, β tubulin were".
  7. Line 143 - 145: This sentence is confusing. Does this human polymorphism result in cancer susceptibility in human?
  8. Line 146, "Supporting these clinical and mouse studies," is also confusing, because the studies are described after this sentence.
  9. If chapters 2.1.2 an 2.1.3 refer to the same variant, they may be integrated into one chapter.
  10. Line 246, "PDR" should be "proliferative diabetic retinopathy (PDR)" or "proliferative DR".

Author Response

Reviewer 2

Comment:  In Abstract, italicized LOX and LOXL1 - 4 means the human genes, whereas in Introduction, these genes are described as italicized Lox and Loxl1 - 4 that means mouse genes.

Response: We have corrected nomenclature as pointed out, also in response to Reviewer 1’s similar comment.

Comment: Line 36, "in other proteins, [5,6]": this comma is not necessary.

Response: Thank you for pointing this out. This was corrected.

Comment: Line 60, "Lox expression" should be "LOX expression".

Response: This has been corrected.

Comment: Line 113, The title of 2.1.1. should be changed because this chapter contains not only CIN85, but also Hsp70 and c-Raf. More broad terms, e.g., "Intracellular target of LOX-PP" may be suitable for the title.

Response: We totally agree and have changed the title of this section. Thank you.

Comment: Line 117, "NAPDH" should be "NADPH".

Response: This has been corrected.

Comment: Line 132, "Hsp70, cRaf, α, β tubulin and were" should be " Hsp70, cRaf, and α, β tubulin were".

Response: This has been corrected.

Comment: Line 143 - 145: This sentence is confusing. Does this human polymorphism result in cancer susceptibility in human?

Response: The effects of this polymorphism on cancer has been demonstrated both in humans and mice. This has now been clarified in the manuscript.

Comment: "Supporting these clinical and mouse studies," is also confusing, because the studies are described after this sentence.

Response: We agree and edited the paragraph to be more precise.

Comment: If chapters 2.1.2 an 2.1.3 refer to the same variant, they may be integrated into one chapter.

Response: We would prefer to keep the FGF/LOX-PP summary separate from the section 2.1.2 because data were obtained in both cancer and non-cancer contexts, and we have integrated these findings for the first time in this review.

Comment: Line 246, "PDR" should be "proliferative diabetic retinopathy (PDR)" or "proliferative DR".

Response: We have removed this abbreviation since it occurred only once.